# Grading Distress of Different Animal Models for Gastrointestinal Diseases Based on Plasma Corticosterone Kinetics

**DOI:** 10.3390/ani9040145

**Published:** 2019-04-03

**Authors:** Simone Kumstel, Guanglin Tang, Xianbin Zhang, Hagen Kerndl, Brigitte Vollmar, Dietmar Zechner

**Affiliations:** Rudolf-Zenker-Institute of Experimental Surgery, University Medical Center, 18057 Rostock, Germany; guanglintang@hotmail.com (G.T.); zhangxianbin@hotmail.com (X.Z.); hagen.kerndl@uni-rostock.de (H.K.); brigitte.vollmar@uni-rostock.de (B.V.); dietmar.zechner@uni-rostock.de (D.Z.)

**Keywords:** corticosterone, stress, pancreatitis, liver fibrosis, pancreatic cancer, hypothalamic-pituitary-adrenal axis

## Abstract

**Simple Summary:**

Animal welfare is an important aspect of biomedical research. Many regulations have been implemented to combine high quality of research with minimal harm to laboratory animals. These guidelines also demand a prospective severity assessment of each animal model. A comparison of distress between animal models could allow realistic harm and benefit analysis and an appropriate use of refinement methods. However, studies comparing distress between different animal models are still rare. One good parameter for analyzing distress is the concentration of the stress hormone corticosterone in the blood. Therefore, we compared the corticosterone kinetics of distinct gastrointestinal animal models. The aim of this study was to evaluate which parameter the highest corticosterone concentration or the duration of increased stress hormone level could be used to quantify distress. We observed a significant increase of corticosterone 30 min after stress induction in all animal models. However, the corticosterone kinetics differed between the distinct interventions. Both the absolute value and the duration of increased corticosterone level correlated directly with an assessed distress score. We conclude that both variables of corticosterone kinetics are valid parameters to compare distress between animal models.

**Abstract:**

Comparative studies for evaluating distress in established animal models are still rare. However, this issue is becoming more important as a consequence of worldwide appreciation of animal welfare. One good parameter for evaluating distress is the quantification of corticosterone. We hypothesized that not just the absolute value but also the duration of increased corticosterone concentration in the blood is an important aspect for evaluating animal distress. Therefore, we analyzed plasma corticosterone concentrations 30, 60, 120, and 240 min after induction of pancreatitis by cerulein, liver damage by carbon tetrachloride, liver damage by bile duct ligation, and after orthotopic injection of pancreatic cancer cells. We also evaluated corticosterone kinetics after injection of distinct carrier substances. Compared to phosphate buffered saline, dimethyl sulfoxide leads to dose-dependent higher and longer-lasting circulating corticosterone concentrations. In all disease models, we observed significantly increased corticosterone concentration 30 min after stress induction. However, the corticosterone kinetics differed among the animal models. Both the absolute value of corticosterone concentration and the duration correlated positively with the quantification of animal distress by a score sheet. This suggests that both variables of corticosterone kinetics might provide a solid basis for comparing and grading distress of different animal models.

## 1. Introduction

Animal welfare is important for biomedical research and many regulations have been implemented over the last decades to combine high quality of research with minimal harm to animals. In many countries and especially in the European Union (EU) after the introduction of the EU directive 63/2010, animal-based research should strictly follow the 3R principle (reduce, refine, replace) and the severity level of each animal experiment must be defined and reported by the scientists. In the last decades, many animal models for different diseases have been developed for biomedical research. However, usually only parameters describing the pathology are evaluated and studies comparing the distress between animal models are still rare [1,2].

For the evaluation of distress in laboratory animals, the analysis of plasma corticosterone has proven to be a sensitive method [3,4,5]. However, its validity for comparing distress between different animal models and strains still needs to be explored. Corticosterone is the most important glucocorticoid regulating the stress response in rodents. Different stressors induce the activation of the hypothalamic–pituitary-adrenal (HPA) axis and provoke the release of corticotrophin-releasing hormone (CRH) and arginine vasopressin (AVP) from neurons in the paraventricular nucleus of the hypothalamus (PVN). The transport of these factors to the pituitary gland leads to the secretion of adrenocorticotrophic hormone (ACTH), which stimulates the adrenal cortex to release corticosterone into the circulation [6]. Corticosterone regulates the metabolism to cover the energy supply for the stress response [7] and modulates defense reactions to stress [4]. 

The plasma corticosterone level, however, does not always indicate distress, but is also influenced by physiological processes such as the circadian rhythm [3,8,9], sexual arousal [10], and the estrus cycle [3,11,12]. Nevertheless, it is widely accepted that repeated and unpredictable stressors increase the corticosterone concentration in the blood [3,5,13]. However, the kinetics of corticosterone concentration in the blood were reported to be distinct between different stressors. For example, timing differences in the up- and downregulation of corticosterone in the blood were observed after restraint stress, electric shocks, shaker stress, heat stress, and forced swimming [3,14,15,16]. These studies suggest that an assessment of corticosterone concentration at only one point in time is insufficient, for a conclusively comparison of distress between different stressors. This point is especially valid when comparing diseases in distinct animal models, which are induced by completely different methods, for instance by injection of chemicals or by surgical intervention. The purpose of this study was to analyze the corticosterone kinetics after inducing distinct gastrointestinal diseases. Gastrointestinal diseases are the most common cause of hospital admission and the third most common cause of death [17]. In particular, pancreatic cancer, pancreatitis, and liver fibrosis require new treatment strategies [18,19,20]. Therefore, these diseases are part of many preclinical trials using animal models [21,22,23]. In this explorative study, we assessed the optimal time point to compare the maximal plasma corticosterone concentration after induction of different gastrointestinal animal models using different mouse strains. Additionally, we evaluated which parameter of the corticosterone profile the absolute value or the duration of increased corticosterone concentration is a good indicator of distress. A comparison of distress of these widely-used gastrointestinal animal models could be used to adjust refinement methods in future.

## 2. Materials and Methods 

### 2.1. Study Background, Animals, and Interventions

The current study is part of a DFG (German Research Foundation)-funded multicenter approach (FOR 2591) including 15 different research groups of eight institutions in Germany and Switzerland, involving five different animal species. The overall goal of this research group is to establish a species and animal model severity assessment framework and consequently to minimize severity in laboratory animals by employing improved refinement methods [24]. The current study is therefore an essential first step to analyze and to compare corticosterone kinetics after different interventions and to evaluate the optimal time point to measure corticosterone.

The mice were kept in a room with 12/12 h dark light cycle (light period: 07.00 h–19.00 h) with food and water ad libitum. The mice were kept in groups of 2–5 animals in type III cages and enrichment was provided by paper role and rodent wood. The mice were bred in our own facility under specific pathogen free conditions. All experiments were approved by the Landesamt für Landwirtschaft, Lebensmittelsicherheit und Fischerei Mecklenburg-Vorpommern (7221.3-1-019/15; 7221.3-1-002/17). The decisions of the local authority are in accordance with the Protection of Animals Act (Deutsches Tierschutz Gesetz) and the European Directive 2010/63/EU. For all animal models, male mice were used, to exclude the influence of the estrus cycle on the corticosterone concentration [3,12,13]. During all procedures, the mice were picked up by their tail. For analyzing the distress after injecting distinct solvents, which we usually use as different therapeutic agents, 15–20-week-old C57BL/6J mice were restrained for about 10–15 seconds while injecting once 2.5 µL/g phosphate buffered saline (PBS), and a low volume (1.25 µL/g) or high volume (2.5 µL/g) of 100% dimethyl sulfoxide (DMSO, Merck, Darmstadt, Germany) into the peritoneal cavity.

### 2.2. Chemical Induction of Pancreatitis and Liver Damage 

Pancreatitis was induced in male 17–19 weeks old C57BL/6J mice by three intraperitoneal (i.p.) injections of cerulein (50 µg/kg; Merck, Darmstadt, Germany) diluted in 0.9% sodium chloride, at a rate of one every hour, three times a week up to 35 days (thus Monday, Wednesday and Friday). The blood for the corticosterone analysis was collected after the third cerulein injection on the second day of cerulein administration. This procedure mimics the early phase of chronic pancreatitis [25]. In order to induce liver damage by carbon tetrachloride (CCl4), male mice of the BALB/cANCrl strain were used at the age of 17–20 weeks. This liver fibrosis model is well established in this mouse strain at our institute [26]. Twice a week (Monday and Friday) for up to 42 days, CCl4 (Merck, Darmstadt, Germany) was i.p. injected at a dose of 0.25 µL/g body weight (after 4-fold dilution in corn oil). This procedure mimics the early phase of an animal model for liver fibrosis [26]. The blood for the analysis of corticosterone was taken on the second day of CCl4 injection. Starting on the day before the first injection, analgesia was provided continuously for both animal models by adding 1250 mg/L metamizol (Ratiopharm, Ulm, Germany) to the drinking water until the end of the experiment.

### 2.3. Surgical Induction of Liver Damage and Pancreatic Cancer

The bile duct ligation (BDL) was performed on male BALB/cANCrl mice at the age of 16–20 weeks. The mice were anesthetized with 1.2–2.5% isoflurane and 5 mg/kg carprofen (Rimadyl^®^, Pfizer GmbH Berlin, Germany) was injected subcutaneously (s.c.) for perioperative analgesia. The eyes were kept wet by eye ointment. A midline laparotomy was executed and the common bile duct was ligated three times with 5-0 silk and transected between the two most distal ligations. The peritoneum and the skin were closed separately by 5-0 prolene suture (Johnson & Johnson MEDICAL GmbH, New Brunswick, NJ, USA) and the mice were placed in front of a heating lamp. For this procedure, the mice were anaesthetized for 25–35 minutes. 

For the orthotopic injection of cancer cells, male 17–24-week-old C57BL/6J mice were used, since this model is well established in this mouse strain [21]. The mice were anesthetized with 1.2–2.5 % isoflurane and 5 mg/kg carprofen (Rimadyl^®^, Pfizer GmbH) were applied by s.c. injection for analgesia. The eyes were kept wet by eye ointment. The abdomen of the mice was shaved and disinfected and the abdominal cavity was opened by laparotomy and 5 µL of the cell suspension (murine cell line 6606 PDA, 2.5 × 105/5 µL cells in matrigel) were injected slowly with a 25-µL syringe into the head of the pancreas (Hamilton Syringe, Reno, NV, USA). The pancreas was placed back into the cavity and the peritoneum was closed by a coated 5-0 vicryl suture (Johnson & Johnson MEDICAL GmbH). The skin was sewed with a 5-0 prolene suture (Johnson & Johnson MEDICAL GmbH). The surgery lasted 15–25 min for each of the mice. 

### 2.4. Preparation of Blood Samples, Evaluation of Distress Score, and Analysis of Plasma

The mice were assigned to four groups in a haphazard manner (no randomization was performed). The blood sampling was executed at the indicated time points (30, 60, 120, and 240 min) after the different i.p. injections or the awakening of the mice after the surgical interventions. The blood was collected only once from each mouse. To assess the plasma corticosterone levels without stress induction (pre-values), the blood of each mouse was collected 2–4 weeks before distinct interventions were conducted. The mice were anesthetized with 5% isoflurane and 100–150 µL blood were collected within 3 minutes by retro orbital puncture in EDTA-Tubes (Microvette^®^, SARSTEDT, Nürnbrecht, Germany) to avoid an increase of corticosterone caused by the procedure [27,28]. The blood sampling was always carried out during a time period from 11:00 to 15:30 h, to avoid the circadian increase of the plasma corticosterone, which starts after 16:00 h (2 h before the dark phase) [3,8,9]. After the blood sampling and the surgical procedures, the mice were euthanized directly via cervical dislocation under isoflurane anesthesia to prevent a distress accumulation from both interventions. The tubes were centrifuged (1200× *g*, 10 min, 20 °C) to separate the plasma and the samples were stored at −20 °C. Immediately before blood collection, the distress score was assessed by only one person in a not blinded manner, according to a scoring sheet (Appendix A), which was modified after Paster et al. [29]. 

The corticosterone concentration in the plasma samples was measured in a blinded fashion with the ELISA-Kit (DE 4164, Demeditec Diagnostics GmbH, Erfurt, Germany) following manufacturer instructions. To evaluate acinar cell damage, the activity of lipase in the blood plasma was analyzed in a blinded fashion with a photometer (cobas c111, Roche Diagnostics, Rotkreuz, Switzerland). As parameters for the acute liver damage the plasma activity of the aspartate aminotransferase (AST) and alanine aminotransferase (ALT) was measured with the identical photometer. 

### 2.5. Data Analysis

All data were analyzed with the program SigmaPlot 12.0 (SYSTAT Software Inc., San Jose, USA). All graphs present data as box plots the 10th and 90th percentile as whiskers. The significances of differences were evaluated by Mann–Whitney rank-sum test, followed by Bonferroni correction. Non-parametric tests were used, because of the low power of normality tests at a limited sample size from 3–5 [30]. Differences with *p* ≤ 0.05, divided by the number of comparisons to the pre-value (*p* ≤ 0.0125), were considered to be significant. Correlations were described by calculating the Spearman correlation coefficient (number of samples: 17).

## 3. Results

### 3.1. Injection of Distinct Solvents Provokes Specific Corticosterone Profiles

Injection of PBS (2.5 µL/g) significantly increased the plasma corticosterone concentration at 30 min, followed by a rapid reduction of the stress hormone concentration (Figure 1A). However, after the i.p. injection of a low volume of DMSO (1.25 µL/g), the corticosterone concentration was significantly increased 30 min as well as 60 min after stress induction (Figure 1B). After the administration of a high DMSO volume (2.5 µL/g), we noticed a significantly increased corticosterone concentration 30 min, 60 min, and 120 min after stress induction (Figure 1C). The maximum of corticosterone concentration was detected at 120 min (Figure 1C). Even 4 hours after the injection, the corticosterone level remained slightly higher in comparison to the values without stress induction (Figure 1C). Independent of the corticosterone profile, the administration of all solvents led to a significant increase of this stress hormone 30 min after injection (Figure 1D). 

### 3.2. Intervention-Specific Profiles in Widely Used Gastrointestinal Mouse Models

The stress hormone response was also measured in an animal model for pancreatitis. The corticosterone concentration was significantly increased at 30 min and 60 min after cerulein injection (Figure 2A). The maximum was detected at 30 min followed by a constant reduction of the corticosterone concentration (Figure 2A). As a parameter for pancreatic tissue damage, lipase activity was quantified. The activity of this enzyme was significantly increased at 30 min, 60 min, 120 min, and 240 min (Figure 2B). We also analyzed the corticosterone profile after cancer cell injection into the pancreas in C57BL/6J mice. The corticosterone concentration was significantly increased at 30 min, 60 min, and 120 min after anesthesia (Figure 3A). The circulating stress hormone remained on a similar constant level until 120 min (Figure 3A). To measure tissue damage during cancer cell injection, lipase activity was evaluated at the distinct time points (Figure 3B). A slight increase of plasma lipase activity was detected at 60 min after cancer cell injection. However, all observed differences were not significant (Figure 3B). In BALB/c mice with CCl4-induced liver damage the highest concentration of corticosterone was observed at 30 min (Figure 4A). However, also after 60 min and 120 min the corticosterone concentration was significantly higher when compared to healthy animals (Figure 4A). In order to compare the stress hormone level with parameters for tissue damage, we determined the time course of plasma activity of AST and ALT (Figure 4B,C). Starting 30 min after the CCl4 injection, we observed a significant increase in the activity of both transaminases (Figure 4B,C). A significant increase in the activity of both enzymes was also observed 60, 120, and 240 min after CCl4 injection (Figure 4B,C).

After the surgical induction of liver damage by bile duct ligation, a significantly increased concentration of circulating corticosterone was detected within 30 min after anesthesia (Figure 5A). In contrast to the chemically induced animal models (pancreatitis, liver damage by CCl4), the concentration of the stress hormone remains steady at a high level until 120 minutes. Even after 4 hours, the corticosterone concentration was still significantly higher in comparison to the plasma samples collected without any intervention (Figure 5A). The AST and ALT values, however, rose significantly at 30 min and sharply increased over time (Figure 5B,C). 

### 3.3. Specific Distress Levels Caused by Distinct Gastrointestinal Diseases 

In order to quantify distress, we evaluated the body weight, general health and the spontaneous as well as flight behavior of mice in all disease models as defined by a score sheet (for details on the score sheet see Appendix A). In the animal model for pancreatitis we observed 30 min after the last cerulein injection an abnormal posture (score 3) in one animal. All other mice received a score of 0. This resulted in an average distress score of 0.75 at the 30-min time point, whereas no distress was detected at the 60-min time point (Figure 6A). The injection of 6606PDA cells into the pancreas resulted in a significant increase of distress (score 2) 30 min after anesthesia, since all mice had a ruffled fur (Figure 6B). At 60 min no distress could be scored anymore (Figure 6B). In the animal model for liver fibrosis by CCl4 all mice had ruffled fur at the 30- and 60-min time point (Figure 6C). This resulted in an average distress score of 2 at both time points. The surgical induction of liver damage by BDL resulted in a very high distress score of 5, due to the accumulation of distress indicators including abnormal posture and passive behavior after touch (Figure 6D). At 60 min just abnormal posture (score 3) was still noticed in all mice (Figure 6D).

## 4. Discussion

The present study demonstrates a significant increase of corticosterone concentration within 30 minutes independent of the stressor and the mouse strain (Figure 1, Figure 2, Figure 3, Figure 4 and Figure 5). However, some interventions, such as injection of cancer cells into the pancreas, provoke a long lasting corticosterone response (Figure 5). Other interventions, such as the injection of cerulein, lead to a transient corticosterone response (Figure 2). Interestingly, not only the absolute value of corticosterone concentration but also the duration of the corticosterone response correlates well with the quantification of animal distress by a score sheet (Table 1).

The i.p. injection of different carrier substances is widely used in animal experiments for chemical disease induction and therapeutic intervention. Nevertheless, this intervention represents an important stressor, measurable by an increase of the stress hormone corticosterone. To evaluate the distress of the most important carrier substances of water- and lipo-soluble drugs, corticosterone kinetics after PBS and DMSO injection have been analyzed. A transient corticosterone response was noticed after PBS injection (Figure 1A). Other studies also demonstrated a quick downregulation of the stress hormone within 60 min [31,32]. This quick response is initiated by corticosterone itself due to a receptor-mediated negative feedback inhibition of the precursor hormones CRH and ACTH in the neurons of the PVN [33,34,35]. However, it is not possible to clearly differentiate between distress caused by handling of the animals and distress caused by the injection. The distress of only handling animals also leads to a significant increase in blood corticosterone concentration within 15 min, followed by a quick decrease, similar to observations made after PBS injection [36]. This suggests that PBS injection and handling of mice cause similar levels of distress. After DMSO administration, a longer-lasting corticosterone response in comparison to PBS injection was detected. Some studies suggest that DMSO can cause a longer-lasting activation of the HPA axis, independent of the nerval innervations of the hypothalamus [37,38]. We refrained from comparing the i.p. injections of PBS and DMSO directly with the induction of gastrointestinal disease, because the application of analgesia might have influenced the corticosterone level.

Intervention-specific corticosterone characteristics were analyzed after chemical and surgical disease inductions in liver and pancreas. Acute liver damage was induced in the murine liver of BALB/c mice by chemical induction with CCl4 and surgical intervention by BDL. Both interventions caused a significant increase of corticosterone concentration within 30 min. However, the injection of CCl4 caused a transient increase in corticosterone concentration, whereas the BDL provoked a significantly increased corticosterone concentration for a longer period (cf. Figure 4 and Figure 5). Similar differences were observed when analyzing animal models, which focus on pancreatic diseases in the C57BL/6J strain. The injection of cerulein caused a transient increase in corticosterone, whereas the surgical induction of pancreatic cancer by laparotomy and cell injection in the pancreas caused a significant increase in corticosterone concentration for a longer period (cf. Figure 2 and Figure 3). Even so, it is widely accepted that the absolute corticosterone concentration differs among distinct mice strains [5,15,32], the above-mentioned characteristic kinetics after chemical and surgical disease induction could be observed in both strains. A long-lasting corticosterone response for 2–6 hours after surgical interventions like vasectomy and catheterization surgery was also observed in other studies [39,40]. On the contrary, after nonsurgical interventions such as nicotine injection [32] or restrain stress [27,41,42] the corticosterone concentration usually declines within 60 to 120 min. The results indicate that the long-lasting corticosterone response might be a characteristic feature for the stress response after surgical procedures. 

Several reasons might account for these altered corticosterone kinetics. One reason could be the effect of anesthesia. Indeed, other studies reported a significant increase in corticosterone concentration within minutes of anesthesia using isoflurane. However, the increased corticosterone concentration was transient and declined after 60 min. Therefore, anesthesia alone is probably not responsible for the long-lasting activation of the HPA axis. Another reason might be that the summation of stressors such as pain, wound tension, tissue damage and the release of cytokines provokes a long lasting glucocorticoid response [43]. Moreover, the cytokine interleukin-6 (IL-6) can influence the HPA axis directly by stimulating the glucocorticoid release on the adrenal glands [43,44]. The release of IL-6 usually starts within 60 min after surgical procedures [45,46] and positively correlates with the corticosterone concentration [43,47]. In conclusion, the long-lasting corticosterone response after surgery might arise from the summation of distress parameters that influence the HPA axis. 

One aim of this study was to evaluate which aspect can be used to judge animal distress: the maximal corticosterone concentration or rather the period of time for the significantly increased corticosterone response. Quantifying distress by our score sheet allowed us to rank the disease from highest distress (score 5) after BDL to lowest distress (0.75) during pancreatitis (Table 1). When comparing corticosterone concentration at 30 minutes between these animal models, a very similar rank order could be observed (Table 1). Indeed, the corticosterone concentrations of all animals, which were evaluated in all four animal models, correlated well with the distress score (Spearman’s rank correlation coefficient: 0.67; *p* = 0.003). Previous studies support the hypothesis of a direct correlation between corticosterone concentration and stressor intensity [32,48,49]. Physical stress indicators, such as body weight loss, increased heart rate, and decreased food intake are also known to be associated with increased plasma corticosterone concentration. This emphasizes the relevance of this parameter for the analysis of distress [14,31,42]. However, since corticosterone is also influenced by other physical responses, i.e., circadian rhythm [3,8,9], estrus cycle [3,11,12], or sexual arousal [10], this parameter should always be assessed together with other stress or welfare parameters like distress score or body weight.

Furthermore, when comparing the duration of significantly increased corticosterone concentration between the four animal models a rank order could be established (Table 1). This rank order was identical to the rank order, when we scored distress (Table 1). This suggests that also the duration of increased corticosterone concentration is a relevant parameter for the analysis of distress. This conclusion is supported by various studies. For example, preventive wheel running shortened the glucocorticoid response after acute restrain stress and also reduced anxiety-like behavior [50]. Moreover, buprenorphine treatment decreases the duration as well as the corticosterone concentration after catheterization [40]. Therefore, we suggest that both parameters of corticosterone kinetics, absolute value of corticosterone concentration as well as duration of corticosterone response, are appropriate parameters for quantifying distress. Additionally, our findings indicate that corticosterone kinetics can be used as an important parameter for the grading of distress level between distinct animal models. Though the absolute value and duration of corticosterone kinetics correlate well with the assessed distress scoring, we want to mention that a direct comparison of two distinct mouse strains should be handled with care, as strain might act as a confounding variable. For example, BALB/c mice are known to show higher absolute corticosterone values and a longer duration compared to the C57BL/6J strain, which is probably caused by higher anxiety [5] or genetic alterations in the HPA axis [15]. Nevertheless, we got very similar results in both mouse strains after different interventions: Not only the absolute value of corticosterone concentration after 30 minutes but also the duration of the corticosterone response correlates well with the quantification of animal distress by a score sheet. We hope that the purposely increased heterogeneity in the experimental design improves external validity [51,52] and that these results will, therefore, provide an important basis to compare corticosterone concentrations of different animal models across mouse strains.

It is widely published that the time point of highest corticosterone concentration is dependent on the kind of intervention. Some studies noticed the peak of corticosterone response 20–30 min after the injection of toxic chemicals [16,32], or restrain stress [27,42]. However, after some other stressors like electric shocks [53] or acute shaker stress [14], the maximum of corticosterone concentration can be observed 60 min after distress. Thus, for comparing animal models, it is advisable to evaluate corticosterone kinetics previously on a few animals instead of relying on only one random time point. This procedure subsequently allows the reduction of animals. However, our data suggest that the corticosterone concentration at 30 minutes is a suitable parameter for comparing distress of our gastrointestinal disease models.

## 5. Conclusions

The present study demonstrated that distress caused by different interventions correlates with the peak and duration of corticosterone kinetics in distinct gastrointestinal mouse models. These results support the use of corticosterone kinetics as an important readout parameter to grade distress in animal models.

## Figures and Tables

**Figure 1 animals-09-00145-f001:**
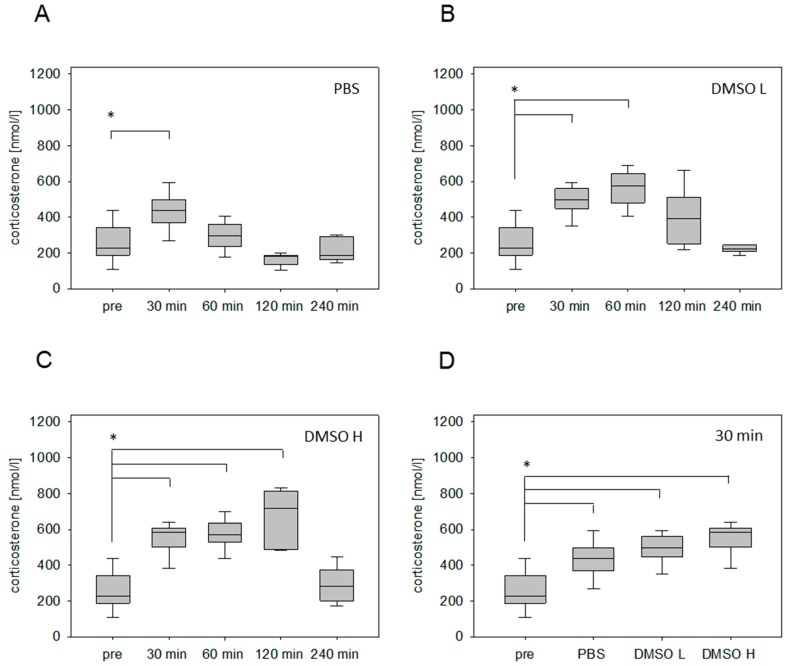
Corticosterone concentration in blood plasma after intraperitoneal (i.p.) injection of different solvents. Time-dependent glucocorticoid response before (pre) and after injection of phosphate buffered saline (PBS) (2.5 µL/g) (**A**), low volume of dimethyl sulfoxide (DMSO L: 1.25 µL/g) (**B**) and high volume of DMSO (DMSO H: 2.5 µL/g) (**C**). Comparison of corticosterone concentration measured 30 min after injection of indicated solvents (**D**). Significant differences: * *p* ≤ 0.0125; *n* = 20 (pre); *n* = 5 (30, 60, 120, 240 min).

**Figure 2 animals-09-00145-f002:**
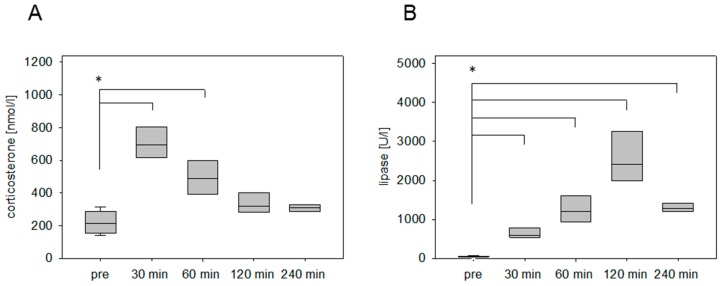
Corticosterone concentration and lipase activity after the induction of pancreatitis. Analysis of time-dependent corticosterone release before (pre) and after i.p. injection of cerulein (50 µg/kg) (**A**), as well as evaluation of lipase activity (**B**). Significant differences: * *p* ≤ 0.0125; *n* = 16 (pre); *n* = 4 (30, 60, 120, 240 min).

**Figure 3 animals-09-00145-f003:**
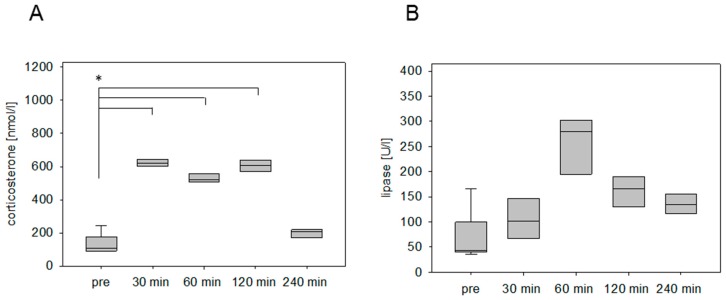
Corticosterone concentration after laparotomy and cancer cell injection into the pancreas. Evaluation of stress hormone release (**A**) as well as lipase activity (**B**) before surgery (pre) and at the indicated time points after cessation of anesthesia. Significant differences: * *p* ≤ 0.0125; *n* = 8 (pre); *n* = 4 (30, 60, 120, 240 min).

**Figure 4 animals-09-00145-f004:**
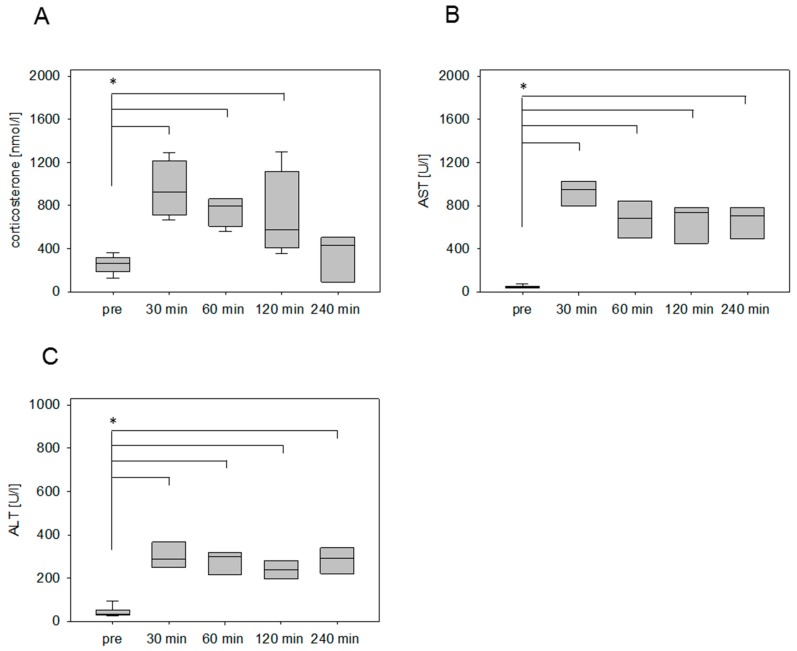
Corticosterone concentration and transaminase activities after chemical induction of liver damage. Analysis of stress hormone release of mice before (pre) and at the indicated time points after the i.p. injection of carbon tetrachloride (CCl4; 0.25 µL/g) (**A**) as well as the evaluation of aspartate aminotransferase (AST) (**B**) and alanine aminotransferase (ALT) activity (**C**). Significant differences: * *p* ≤ 0.0125; *n* = 15 (pre), *n* = 3–4 (30, 60, 120, 240 min).

**Figure 5 animals-09-00145-f005:**
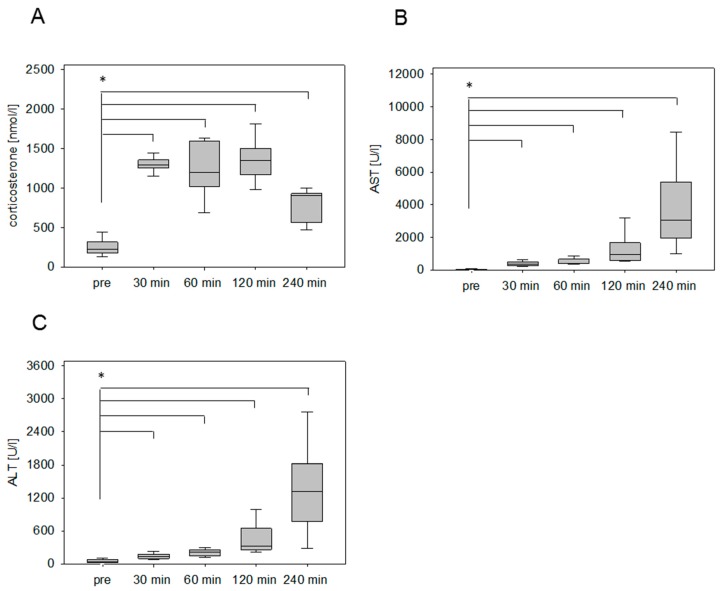
Stress hormone release and transaminase activities after surgical induction of liver damage by bile duct ligation (BDL). Evaluation of corticosterone concentration of mice before (pre) and after the awakening of the mice from anesthesia (**A**), as well as the evaluation of transaminase activity of AST (**B**) and ALT (C) as parameters for the tissue damage. Significant differences: **p* ≤ 0.0125; *n* = 20 (pre); *n* = 5 (30, 60, 120, 240 min).

**Figure 6 animals-09-00145-f006:**
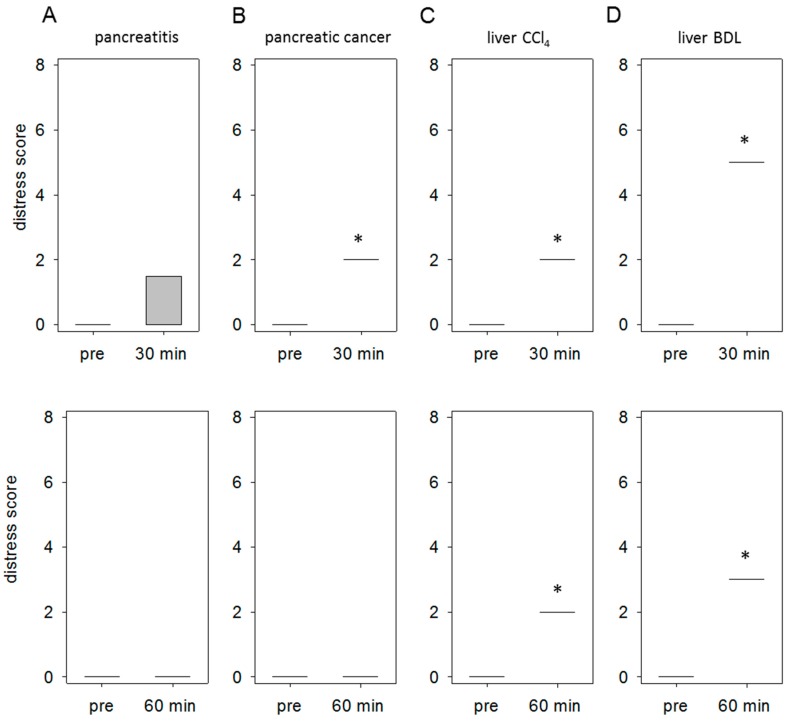
Comparison of distress between distinct mouse models. Comparison of the distress score at 30 min and 60 min after injection or end of anesthesia to the score taken before any intervention (pre). Thirty minutes after the induction of pancreatitis an abnormal posture was observed in only one animal, which results in an average distress score of 0.75 (**A**). Tumor cell injection into the pancreas resulted in a score of 2 (ruffled fur) in all mice at 30 min (**B**). Induction of liver damage by CCl4 led to a score of 2 (all animals displays ruffled fur) at 30 as well as at 60 min (**C**). Bile duct ligation resulted in a score of 5 at 30 min (due to abnormal posture and passive behavior after touch) and a score of 3 at 60 min (abnormal posture) (**D**). Significant differences: * *p* ≤ 0.05; *n* = 4–5 (pre); *n* = 4–5 (30, 60 min).

**Table 1 animals-09-00145-t001:** Animal models ranked according to distress, as well as absolute value and duration of corticosterone concentration. The mouse strains used for each animal model are specified. BDL: bile duct ligation.

BALB/c			C57BL/6J			
**Distress** (mean value of distress scored at 30 min)						
BDL	>	CCl_4_	=	tumor cell injection	>	pancreatitis
5.0	>	2.0	=	2.0	>	0.75
**Corticosterone concentration at 30 min** (interquartile range 25–75% in nmol/L)						
BDL	>	CCl_4_	>	tumor cell injection	~	pancreatitis
1218.1–1383.4	>	710.6–1215.1	>	599.0–649.6	~	602.4–831.6
**Duration of corticosterone response** (latest time point in minutes with a significant increase in corticosterone concentration)						
BDL	>	CCl_4_	=	tumor cell injection	>	pancreatitis
240	>	120	=	120	>	60

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
