# Peer review of "Grading Distress of Different Animal Models for Gastrointestinal Diseases Based on Plasma Corticosterone Kinetics"

_animals, 2019, doi:10.3390/ani9040145_

Round 1
Reviewer 1 Report
I believe the authors have addressed my main concerns in an adequate manner, and therefore do not oppose the publication of this article.
Author Response
Dear Reviewer,
Thank you very much for approving our manuscript.
Reviewer 2 Report
Thank you for addressing my comments. I feel that the experimental design could be improved questioning the validity of the results. Additionally, I feel that the paper does not clarify the intention of the experiments, that is to compare animal models of GI disease that are equivalent. If the focus of the paper was to compare the welfare implications of animal models that answer the same questions it would be more valuable. The fact that strain is not taken into account as a factor is a great flaw of the experimental design making difficult to compare the different animal models. Additionally, it is not made clear whether the induction of disease in these experiments were done for the sole purpose of comparing corticosterone measures or if they were done parallel to other studies to gather further information. Finally, I'm concerned about the lack of refinement for handling methods and for blood sampling
Author Response
Comments to Reviewer 2
Dear Reviewer,
Thank you very much for your constructive suggestions, regarding the comprehension of our study background and design. Attached you will find the point to point reply to your comments. Additionally, we changed the manuscript to state our study background more precisely. We hope to be able to convey our research interest regarding this explorative study and its implication for following projects of our group.
Comment 1:
The paper does not clarify the intention of the experiments, that is to compare animal models of GI disease that are equivalent. If the focus of the paper was to compare the welfare implications of animal models that answer the same questions it would be more valuable.
Answer to comment 1:
We thank the reviewer for the helpful advice and like to give some background thoughts for a better understanding of the intention of our study:
Background:
The current study is part of a DFG (German Research Foundation)-funded multicenter approach (FOR 2591) including 15 different research groups of 8 institutions in Germany and Switzerland, involving 5 different animal species. The overall goal of this research group is to establish a severity assessment framework and consequently to minimize distress in laboratory animals (Bleich et al., 2017). Valid distress parameters are necessary for comparing the severity of different animal models. Plasma corticosterone is one important parameter and its validity to compare distress between different animal models, strains and even species still needs to be explored. The final aim of this severity assessment framework is a species and model overarching grading system for severity level.
Intentions:
This is the reason why we purposely used different mouse strains for our study and why we used different animal model of GI disease. However, we asked in these different models the same question, which was as follows: what is the optimal time point to assess the maximal plasma corticosterone concentration in order to compare the animal distress after different procedures (or induction of different diseases)? and, which aspect of the corticosterone profile, either the absolute value or the duration of increased corticosterone, is a good indicator of distress? Additionally, the use of different animal models increases the heterogeneity in the experimental design. Conclusions based on heterogeneous study designs are often more robust than studies without a heterogeneous study design (Richter et al., 2009; Voelkl et al., 2018).
These aspects have now been better addressed in the revised version (see section introduction, material & methods, lines 52-53, 76-78 and 84-90, highlighted in yellow). In order to stress this point we also slightly changed the title (see line 2, highlighted in yellow).
Bleich, A.; Tolba, R. H. How can we assess their suffering? German research consortium aims at defining a severity assessment framework for laboratory animals. Laboratory animals. 2017, 51, 667, doi:10.1177/0023677217733010. PMID:29160175.
Richter, S. H.; Garner, J. P.; Würbel, H. Environmental standardization: cure or cause of poor reproducibility in animal experiments? Nature Methods. 2009, 6 (4): 257-261.
Voelkl, B.; Vogt, L.; Sena, E. S.; Würbel, H. Reproducibility of preclinical animal research improves with heterogeneity of study samples. Plos Biology. 2018, 16 (2): e2003693. https://doi.org/10.1371/journal.pbio.2003693
Comment 2:
The fact that strain is not taken into account as a factor is a great flaw of the experimental design making difficult to compare the different animal models.
Answer to comment 2:
As stated in our comment to your point #1, the overall goal of the DFG research group FOR 2591 is to establish a severity assessment framework for comparison of distress between different animal models, strains and even species. Thus, the inclusion of different mouse strains is important. In both mouse strains, we observed the same characteristics in the corticosterone kinetics upon chemical and surgical diseases induction. And even the significant increase of corticosterone after 30 min was not strain specific. Thus, these results might provide an important basis for a reliable comparison of distress between animal models in the future.
We are aware of the strain as confounding variable and have this already discussed in the original manuscript. However, with the goal of the research group in mind aiming at the establishment of most robust parameters for severity assessment, we purposely increased heterogeneity of the experimental design by including different mouse strains. In theory, this strategy may reduce internal validity by reduction of standardization and increase of data variation. In turn, this strategy theoretically improves external validity (Vissers et al., 2001) and thus, generalization of scientific findings. With view on the current world-wide debate on the “replication crisis”, the strategy to increase heterogeneity of the experimental design and thus the robustness of data and conclusions is recommended and starts to be commonly accepted (Richter et al., 2009).
These aspects have now been better addressed in the revised version of the manuscript (lines 52-53, 84-90 and 354-359, highlighted in yellow).
Vissers, G.; Heyne, G.; Peters, V.; Guerts, J. The Validity of Laboratory Research in Social and Behavioral Sciences. Quality and Quantitiy. 2001, 35: 129-145.
Richter, S. H.; Garner, J. P.; Würbel, H. Environmental standardization: cure or cause of poor reproducibility in animal experiments? Nature Methods. 2009, 6 (4): 257-261.
Comment 3:
It is not made clear whether the induction of disease in these experiments were done for the sole purpose of comparing corticosterone measures or if they were done parallel to other studies to gather further information.
Answer to comment 3:
The animal experiments were performed as described in the present manuscript. The animals did not receive any other additional interventions and, thus, did not serve in conjunction with another study.
This project has to be seen as an explorative study with the purpose to evaluate the optimal time point to measure corticosterone for comparison of distress between the animal models used. In addition, the intention was to analyze the corticosterone kinetics after different disease inductions and their implication for distress quantification. The present study is completed in itself and has two conclusive messages: (1) 30 minutes is the optimal time point to measure corticosterone to compare distress between different gastrointestinal animal models and (2) both the absolute value and the duration of increased corticosterone concentration are important parameters to quantify distress.
However, the results of this study provide an important basis for subsequent four studies (currently under revision, in submission and in preparation; Kumstel et al., Abdelrahman et al. and Tang et al. see below), in which we performed extensive distress analysis along with therapeutic approaches in the mentioned gastrointestinal animal models. By doing this, we applied a variety of behavioral, biochemical and physiological distress parameter as well as innovative statistical models to grade severity levels and to optimize analgesia regimens.
Kumstel, S.; Wendt, E. H. U.; Eichberg, J.; Talbot, S. R.; Häger, C.; Zhang, X.; Schönrogge, M.; Palme, R.; Bleich, A.; Vollmar, B. and Zechner, D. Grading animal distress in a simple and non-invasive manner to define side effects of drugs and well-being of animals. Under review in Scientific reports (03.2019).
Kumstel, S.; Vasudevan, P.; Palme, R.; Zhang, X.; Wendt, E.H.U.; Vollmar, B. and Zechner, D. Benefits of non-invasive methods for distress analysis compared to telemetry (In preparation).
Abdelrahman, A.; Kumstel, S.; Liebig, M.; Wendt, E.H.U.; Eichberg, J., Palme, R., Thum, T., Vollmar, B. and Zechner, D. A novel multi-parametric analysis of non-invasive methods to assess animal distress during chronic pancreatitis. Submitted in Scientific reports (03.2019).
Tang, G.; Seume, N.; Zhang, X.; Abshagen, K.; Kumstel, S.; Vollmar, B. and Zechner, D. Bile duct ligation causes more distress than CCl4 induced liver damage: A comparison of two multiparametric analysis methods (In preparation).
Both the current study as well as the four above mentioned studies were performed not in conjunction to each other, however were ethically approved in two animal application by the local authority in case of BDL, CCl4 and pancreatitis: no.: 7221.3-1-002/17 and in case of pancreatic cancer and i.p. injections: no.: 7221.3-1-019/15. This is now better addressed in the revised version of the manuscript (see section material & methods, line 84-90, highlighted in yellow).
Comment 4:
I’m concerned about the lack of refinement for handling methods and for blood sampling.
Answer to comment 4:
The methods are described in the manuscript, as they have been performed.
The advantage of retro orbital puncture instead of other blood sampling method is that it can be performed very quickly (100 μl, 10 sec) without the need of subsequent manual hemostasis. A quick blood sampling within 3 minutes including anesthesia is mandatory for corticosterone measurement to exclude an influence of the sampling method to the corticosterone level (Vahl et al., 2005; Kugler et al.,1988). Retro orbital puncture is more than 7 times faster than saphenouse vein puncture and more than 15 times faster than tail vein puncture (van Herck et al., 2001). We added refinement to this procedure by a fast anesthesia via 5 % isoflurane within 2 minutes. This leads to a fast and deep narcosis for 40-60 seconds. The blood sampling itself takes 10-20 seconds, performed by an experienced researcher.
However, being permanently on the way to better imply the 3R principle, we are currently establishing alternative methods in both handling (cup and tunnel handling) and blood sampling (facial vein) and will routinely apply these methods in the future.
Vahl, T.P.; Ulrich-Lai, Y.M.; Ostrander, M.M.; Dolgas, C.M.; Elfers, E.E.; Seeley, R.J.; D'Alessio, D.A.; Herman, J.P. Comparative analysis of ACTH and corticosterone sampling methods in rats. American journal of physiology. Endocrinology and metabolism. 2005, 289, E823-8, doi:10.1152 /ajpendo.00122.2005.
Kugler, J.; Lange, K.W.; Kalveram, K.T. Influence of bleeding order on plasma corticosterone concentration in the mouse. Experimental and clinical endocrinology. 1988, 91, 241–243, doi:10.1055/s-0029-1210754.
van Herck, H.; Baumans, V.; Brandt, C.J.; Boere, H.A.; Hesp, A.P.; van Lith, H.A.; Schurink, M.; Beynen, A.C. Blood sampling from the retro-orbital plexus, the saphenous vein and the tail vein in rats: Comparative effects on selected behavioural and blood variables. Laboratory animals. 2001,35, 131–139, doi:10.1258/0023677011911499.
Round 2
Reviewer 2 Report
Dear Authors,
Many thanks for addressing my comments and concerns. Following your responses I will now approve the manuscript for publication.
I do encourage you however, to do these type of studies in conjunction with scientists that are already doing the same procedures to avoid the unnecessary use of more animals.
Kind regards
This manuscript is a resubmission of an earlier submission. The following is a list of the peer review reports and author responses from that submission.
Round 1
Reviewer 1 Report
This study makes a comparison between corticosterone kinetics – mainly increase and duration of high levels – following different stressors, some of which known chemical inducers of gastrointestinal animal models of disease. The authors make the case that increase and duration can be an indicator of distress, and that this can inform of the severity of the procedure, with particular relevance for animal research in the EU, under 63/2010/EU Directive.
There is some merit to this approach, considering that distress is known to be a factor of both intensity and duration of stressors. It is also relevant that these parameters were validated against a previously published clinical score sheet. However, there are issues regarding how the study was designed, carried out interpreted, which should be addressed.
1. In the first paragraph of the discussion, authors state results suggest that corticosterone increase is independent of stressor and mouse strain. I do not think the latter claim can be ascertained, since strain was not a factor in any of the experiments. For it to be a factor that would allow assessing the effect of strain, or lack thereof, the output for the same stressor, from animals of different strains should have been compared. This was not the case.
2. Also, this raises the question of why two mouse strains were used in the first place, since a comparison between corticosterone kinetics of different stressors cannot be made between animal of different strains, as strain becomes a confounding factor. This would call for more experiments to be made, or at the very least the explicit acknowledgment of this serious limitation for the generalizability of results, stemming from a flaw in the experimental design.
3. There is no report of allocation concealment when collecting behavioural data, nor of blind outcome assessment. This has implications to the reliability of the results and therefore it should be explicitly stated that these bias avoidance measures were not in place, and ideally a justification as to why this was the case.
4. Regarding the use of the clinical score sheet, it is not mentioned how many persons (one, two, more?) were in charge of this evaluation.
5. The discussion should also address the fact that blood collection itself is a stressor that should be taken into consideration, when analysing results. While its expected to be a random effect, since it is present for all treatment groups, absolute values are likely to be influenced by sampling method, proficiency of handler, and refinement measures, and differences in blood sampling across laboratories might affect comparability.
6. I would also like to read a more clear description of the models, what humans diseases they intend to mimic, and why these models were selected, in the introduction.
7. The authors make the case that both rise in corticosterone and duration can be used to assess the impact on animals, but do not provide any measure of how both these two parameters can be assessed together and – also together – be used as a single indicator. It is hence not clear whether these two parameters should be used as a standalone estimator, or in combination. There is not much novelty in using just absolute levels, but some gain could be made by relating increased values with duration, for a comparison between different models.
8. While I appreciate refinement, the adding of analgesia (as reported in line 96), should be discussed in terms of the extent to which it might have affected results, especially if it was not provided to all animals in other treatment groups
Minor issues and suggestions. Consider replacing:
Line 2 – “for” by “of”
Line 9 – “for” by “of”
Line 13 – “studies which compare” by “studies comparing”
Lines 20-21 – consider replacing “Both read out parameters, the absolute value as well as the duration of increased corticosterone level, correlated…” by “both readout parameters – the absolute value and duration of increased corticosterone – correlated.
Line 32-33 “dose dependent” by “dose-dependent” and “longer lasting” by “longer-lasting”
Line 35 – “differ” by “differed”
Line 36 – “as well” by “and”
Line 46 “defined by” to “defined and reported by”
Line 47 – remove “the”
Line 49 – “which compare “ by “comparing”
Line 50 – “proved” by “has proven”
Line 60 – “circadiane” by “circadian”
Line 62 – Short-term distress is an oxymoron. If it is short term it is likely not distress, but rather stress If a short-term but very severe stressor causes lasting effects, these become long-term. Consider replacing for “short-term stress” (not “short-time”) .
Line 65 – “restrain” by “restraint”
Line 68 – “distinct” by “different”
Line 71 – See comment 6, above. Also, remove comma after “evaluate”
Line 77 – ad libitum should be in italic
Line 138 – Please indicate why non-parametric tests were used.
Author Response
Please see PDF-File im attachment.

Reviewer 2 Report
Dear Authors,
I have strong ethical concerns about the justification of the work described in this paper. I believe that using naïve mice to induce severe GI models of disease to measure plasma coricosterone is not justified and could have been done by collaborating with colleagues that were currently performing these experiments. Additionally, the paper fails to address the ARRIVE guidelines in several aspects including experimental design (sample size, randomization, experimental unit, controls, etc), description of the animals (e.g. choice of strain, origin of the animals) and husbandry and housing conditions (e.g. type of cages, group housing, etc). Furthermore, some of the practices fail to comply with the Directive's aim of Refinement, such as tail handling and the use of the retro-orbital sinus under anaesthesia for blood sampling.
See attached a pdf with my specific comments
BW

Author Response
Please see PDF-file im attachment.
